# First-line targ veted therapies of advanced hepatocellular carcinoma: A Bayesian network analysis of randomized controlled trials

Wei Ding[1,2©], Yulin Tan[1,2©], Yan Qian[3], Wenbo Xue[1,2], Yibo Wang[1,2], Peng Jiang[1,2], Xuezhong Xu [1,2]*

**1** Department of General Surgery, Wujin Hospital Affiliated with Jiangsu University, Changzhou, China,
**2** The Wujin Clinical College of Xuzhou Medical University, Changzhou, China, **3** Department of Respiration, Changzhou Second People's Hospital Affiliated to Nanjing Medical University, Changzhou, China

© These authors contributed equally to this work.
* xxzdoctor@163.com

**Data Availability Statement:** All relevant data are within the manuscript and its Supporting Information files.

## Abstract

### Purpose

A variety of targeted drug were developed and proved effective and safe in clinical trials. Our study aims to compare the efficacies and safety of different targeted drugs in advanced hepatocellular carcinoma (HCC) for first-line treatment using a Bayesian network meta-analysis approach.

### Methods

PubMed, Embase, and Cochrane library were searched for randomized controlled trials (RCTs) of advanced HCC patients that treated with different targeted drugs. Time to progress (TTP), overall survival (OS) and progress-free survival (PFS) were calculated as hazard ratios (HRs). Objective response rate (ORR) and the proportion of Grade 3–5 adverse events (G3-5AE) were expressed as odds ratios (ORs). We pooled study-specific HRs and ORs using Bayesian network meta-analyses, and ranked first-line drugs by the surface under the cumulative ranking curve (SUCRA).

### Results

A total of 22 RCTs with 9288 patients were enrolled. Brivanib, linifanib, lenvatinib and sorafenib showed a significant improvement on TTP compared to placebo (HR range, 0.45–0.72). Sunitinib (HR = 1.99) and nintedanib (HR = 2.17) showed a significant decline on TTP compared to lenvatinib. Vandetanib (HR = 0.44) and sorafenib (HR = 0.73) showed a significant improvement on OS compared to placebo. There was no significant difference in PFS, ORR and G3-5AE across different drugs. According to cluster rank analysis, vandetanib was the drug with both more effective (OS) and more secure (G3-5AE) compared to Sor followed by nintedanib.

**Funding:** This work was supported by the Science and Technology Bureau of Changzhou Municipal Wujin District (WS201515). The funds are used to pay for layout fees. No author received salary from the funders. The funders had no role in study design, data collection and analysis, decision to publish, or preparation of the manuscript

**Competing interests:** The authors have declared that no competing interests exist.

## Conclusions

This network meta-analysis shows that vandetanib, linifanib, lenvatinib and nintedanib potentially may be the best substitution of sorafenib against advanced HCC as first-line targeted drugs. Vandetanib seems to be the best choise with low quality of evidence. For better survival, novel targeted treatment options for HCC are sorely needed.

## Introduction

An estimated 42,220 new cases and 30,200 new deaths of hepatocellular and intrahepatic bile duct cancers occurred in the U.S. in 2018 [1]. The majority of these deaths are due to hepatocellular carcinoma (HCC), the most common primary hepatic cancer [2]. Globally liver cancer is the fourth causes of cancer death for mortality [3]. HCC is most commonly associated with chronic hepatitis B virus or hepatitis C virus infections, especially with cirrhosis, which limits the feasibility of surgical resection [4]. Liver transplantation and surgical resection still remain the most effective treatment for early stage HCC in good surgical candidates. Unfortunately, the vast majority of patients are in advanced stages with unresectable tumors when they were diagnosed as HCC. In the past, the prognosis of advanced HCC was poor and its treatment was limited to transarterial chemoembolization, radiofrequency ablation, radiotherapy, and systemic pharmacotherapy [5].

In the European SHARP Trial, the multi-targeted small molecule tyrosine kinase inhibitor (TKI) sorafenib was demonstrated to improve median survival over placebo for unresectable HCC patients for the first time [6]. Subsequently, more targeted drugs were developed and proved effective and safe in their phase II or III clinical trials [7]. Although the effectiveness and safety of these drugs have been compared to sorafenib or placebo, they have not been compared to each other head-to-head [8]. In order to further assess the evidence on the efficacy and safety of targeted drugs for the treatment of HCC patients, we performed this Bayesian network meta-analysis (NMA) to compare the survivals, objective response rates (ORRs) and adverse events (AEs) among different targeted drugs for HCC.

## Materials and methods

This review was performed following the preferred reporting items for the systematic reviews incorporating network meta-analyses [9] (S1 File). This network meta-analysis has been registered in the PROSPERO public database (CRD42019145188; http://www.crd.york.ac.uk/PROSPERO).

### Eligibility criteria

We included randomized controlled trials (RCTs) of adult patients with advanced or unresected hepatocellular carcinoma. To avoid the influence of other treatments, the key inclusion criteria for included study populations were as follows: First, it should last more than 4 weeks since most recent local therapy or no local therapy. Second, the patients did not receive prior systemic therapy. The interventions of interest were the targeted drugs for HCC: Bevacizumab plus erlotinib (Bev + Erl), brivanib (Bri), cabozantinib (Cab), codrituzumab (Cod), dovitinib (Dov), erlotinib plus sorafenib (Erl + Sor), everolimus plus sorafenib (Eve + Sor), lenvatinib (Len), linifanib (Lin), nintedanib (Nin), orantinib (Ora), regorafenib (Reg), sorafenib (Sor), sunitinib (Sun), tigatuzumab (Tig), vandetanib (Van). The efficacy and safety outcomes

assessed were time to progress (TTP), overall survival (OS), progress-free survival (PFS), objective response rate (ORR), and the proportion of Grade 3–5 adverse events (G3-5AE).

## Search strategy and study selection

Two researchers (W.D. & Y.T.) systematically searched Pubmed, Embase and the Cochrane Library using a well-developed search strategy without language restriction from inception to Jun 30th, 2019 (S2 Table). Additionally, relevant references were also searched. Unpublished literatures and conference abstracts were not included.

Two reviewers (W.X. & Y.W.) independently screened out the candidate articles via scanning all titles, abstracts and full-texts. A third reviewer (W.D.) made the final decision of the disagreements on candidate articles through consensus.

## Data extraction

Two reviewers (W.D. & Y.T.) extracted relevant data including study author, post time, region, sample size, patient characteristics (age, gender, Eastern Cooperative Oncology Group [ECOG] score, Barcelona Clinic Liver Cancer [BCLC] stage), mode, dose and duration of treatments, and outcomes of interest, independently. A third reviewer (X.X.) made the final decision of the disagreements were via discussion.

## Quality assessment

The quality and the risk of bias of RCTs in this study was assessed using the quality criteria of the Cochrane Collaboration's tool (S1 Table) [10]. The Grading of Recommendations Assessment, Development and Evaluation (GRADE) Working Group approach was used to assess the quality of evidence (QoE) in each of the direct, indirect, and NMA estimates [11, 12]. For direct comparison, we graded evidence from the five aspects; risk of bias, inconsistency, indirectness, imprecision and publication bias, using the standard GRADE approach. For indirect comparison, we rated evidence according to the lower grades of direct comparisons and intransitivity. For NMA estimates, we rated evidence according to the higher grades of the direct and indirect comparisons and incoherence.

## Data synthesis and analysis

Results regarding the OS, PFS and TTP are expressed as hazard ratios (HRs) with 95% confidence intervals (CIs). Results regarding ORR and G3-5AE are expressed as odds ratios (ORs) with 95% CIs. If HRs could not be acquired directly, they were extracted from Kaplan-Meier curves using the method described by Parmar et al [13]. If there were different HRs or ORs based on different evaluation criteria in the same article, we selected the result according to the latest criteria. We did direct pairwise meta-analyses of head-to-head comparisons with RevMan version 5.3.0 (Cochrane Collaboration). The evaluation of heterogeneity among studies was performed by Cochran's Q test and Higgin's $I^2$ statistics. The heterogeneity among all included studies was suggested significant when $I^2 > 50\%$ and/or $P < 0.05$, then a random-effect model was used (DerSimoniane-Laird method); otherwise, a fixed-effect model (Mantel-Haenszel method) was used.

We did Bayesian network meta-analysis with the package 'rjags' version 4–9 and the package 'GeMTC' version 0.8–2 in R version 3.6.1 (https://www.r-project.org). The merged HRs and/or ORs of relative treatment effects are reported as the median and accompanying 95% credibility intervals (95% CrI) of the posterior distribution. We drew network diagrams with Stata/MP version 14.0 (4905 Lakeway Drive, College Station, TX77845, USA). Hierarchical

Bayesian modeling of the present network meta-analysis conformed to the National Institute for Health and Excellence Decision Support Units (NICEDSU) guidelines [14]. To confirm the transitivity and the loop-specific consistency assumption, pairwise direct and indirect effect estimates of closed loops of evidence were inspected for any disagreement [15]. The transitivity was assessed by examining the patient baseline characteristics across studies (age, gender, performance status and tumor stage), treatment stage and treatment protocol [16]. The global test for inconsistency assumption was conducted with the consistency and inconsistency (unrelated mean effects) models. The consistency between direct and indirect comparison was assessed via using a node-splitting test within each network with a loop [17]. The heterogeneity of network meta-analysis was evaluated with the posterior median of the between-trials standard deviation (σ) [14], while comparison-adjusted funnel plot was used to detect the presence of small-study effects or publication bias.

We undertook Markov Chain Monte Carlo (MCMC) simulation as Bayesian inference to calculate the posterior distributions of the interrogated nodes within the framework of the chosen models and likelihood function on the basis of prior assumptions. We used four different sets of initial values to fit the model, yielding 400,000 iterations (100,000 per chain) to obtain the posterior distributions of model parameters then used 50,000 burn-ins and a thinning interval of 10 for each chain. Autocorrelation function was used to assess the convergence of iterations. Global model fit and parsimony was compared between different fitted models to decide on the most accurate model. The posterior mean of the total residual deviance and deviance information criterion (DIC) was used to choose a more appropriate model [18, 19]. The model with a lower DIC was considered as a more appropriate model. The threshold for the statistical significance was chosen as a two-tailed alpha = 0.05.

In order to determinate intervention rankings for outcomes, rank probabilities were extracted from the network meta-analysis. By merging the rank probabilities of different drugs, we generated the surface under the cumulative ranking curve (SUCRA) to simplify the ranking information as a few numbers [20]. It ranks from 0 to 1. It would be 1 when a treatment is certain to be the best and 0 when a treatment is certain to be the worst. To simultaneously compare the efficacy and safety of each drugs, we jointly presented the SUCRA value of OS and G3-5AE on the clustered ranking plot.

## Results

Of 2,808 articles were collected from the databases mentioned above. After removing all duplicate articles and checking all titles and abstracts, 26 studies remained. After further full-texts screening, four researches were excluded (one study [21] was lack of control group, three studies [22–24] were the Sub-studies for previous trials). Finally, a total of 22 RCTs including 9288 patients from all over the world were included in this network meta-analysis (Fig 1) [6, 25–45].

### Study characteristics

The main characteristics of the included studies were summarized in Table 1. The median age in the 22 RCTs ranged from 51 to 70 years with a majority of male participants. The sample size ranged from 67 to 1155 patients. The majority of ECOG scores were 0–1. The majority of BCLC stages were B-C. The included RCTs compared thirteen different drugs (bevacizumab, erlotinib, brivanib, dovitinib, erlotinib, everolimus, lenvatinib, linifanib, nintedanib, orantinib, sorafenib, sunitinib, tigatuzumab, vandetanib), which were only compared to sorafenib or placebo. The targeted drug treatment programs and their abbreviations are shown in S4 file. The main characteristics of the included studies are shown in Table 1. As shown in S1 Table, only twelve studies [25–29, 32, 34, 35, 37, 38, 41, 42] were considered with high risk of bias at

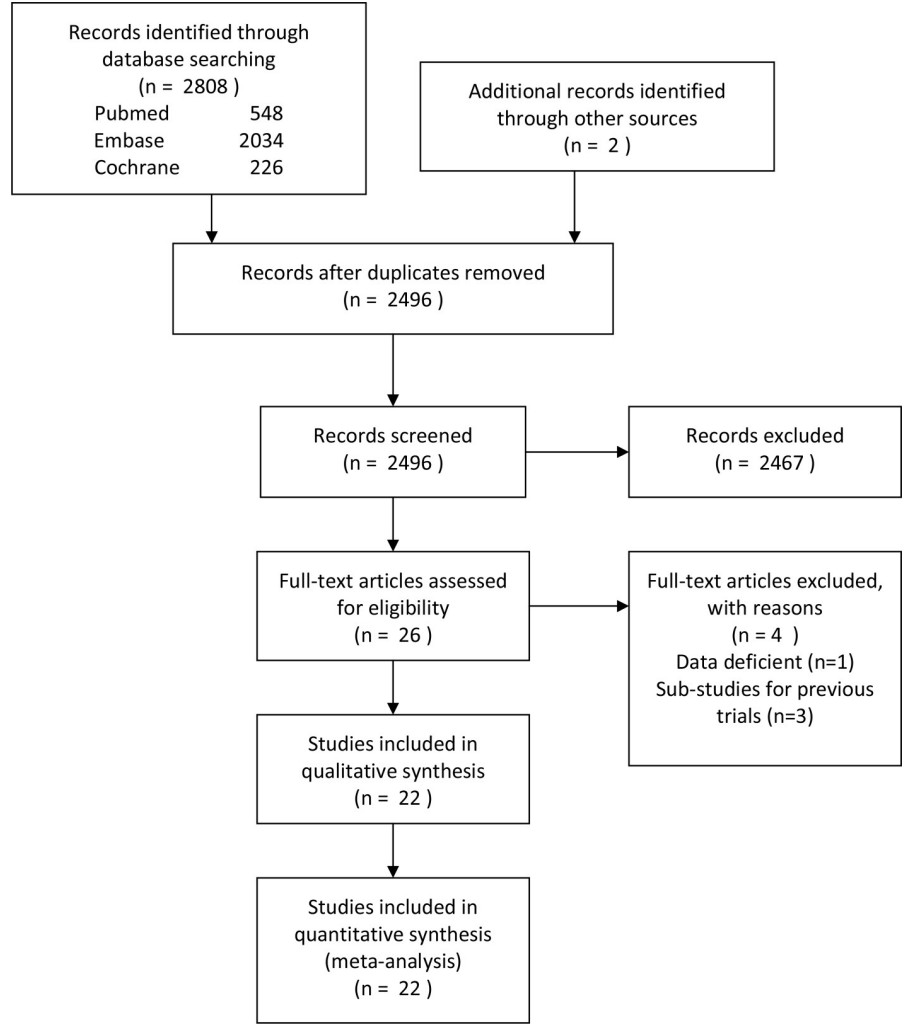

**Fig 1. Flow diagram of study selection.**

blinding of participants and personnel due to their open-label design. There was no evidence of substantial imbalance in the distribution of the effect modifiers across trials in the network. A connected network diagram formed by all evidences is provided in Fig 2. The dosage regimen modes of the same drugs across studies were consistent. By examining the patient baseline characteristics, treatment stage and protocol, there was no evidence that the transitivity assumption was violated in any of the networks.

## Time to progress

Seventeen RCTs [6, 25, 26, 28–31, 33, 34, 36–40, 42, 44, 45] reporting information on TTP were included for meta-analysis. Direct meta-analyses (S1 Fig) confirmed a significant improvement on TTP compared to sorafenib (HR: 0.73; 95%CI: 0.61–0.89) and brivanib (HR: 0.61; 95%CI: 0.48–0.78) over placebo. A connected network diagram formed by TTP is provided in S2 Fig. According to the node-splitting analysis, there was not any significant inconsistency between the direct and indirect comparisons (Pla vs. Bri, $P = 0.54$; Sor vs. Bri, $P = 0.54$; Sor vs. Pla, $P = 0.54$), as shown in S3 Fig. The NMA heterogeneity was low ($\sigma = 0.17$;

**Table 1. Main characteristics of included studies.**

| Study | Year | Intervene | Samples | Age | Gender (M/F) | ECOG (0/1/2) | BCLC stage (A/B/C/D) | HBV infection | White | Black | Asian |
|---|---|---|---|---|---|---|---|---|---|---|---|
| Yen 2018 [25] | 2018 | Nin | 63 | 58 | 57/6 | 35/27/1 | 1/9/53/0 | 40 | 0 | 0 | 63 |
| | | Sor | 32 | 62 | 26/6 | 18/14/0 | 1/1/30/0 | 20 | 0 | 0 | 32 |
| Xu 2018 [26] | 2018 | Sun | 51 | 60 | 42/9 | 29/22/0 | 6/11/34/0 | 41 | 0 | 0 | 51 |
| | | Sor | 53 | 62 | 41/12 | 25/28/0 | 5/16/32/0 | 44 | 0 | 0 | 53 |
| Thomas 2018 [27] | 2018 | Bev + Erl | 47 | 61 | NR | 15/32/0 | 1/14/32/0 | NR | 28 | NR | NR |
| | | Sor | 43 | 61 | NR | 17/25/1 | 4/11/28/0 | NR | 31 | NR | NR |
| Palmer 2018 [28] | 2018 | Nin | 62 | 66 | 48/14 | 32/28/2 | 1/15/45/1 | 4 | 57 | 0 | 0 |
| | | Sor | 31 | 64 | 24/7 | 18/10/3 | 1/7/23/0 | 7 | 24 | 1 | 1 |
| Kudo Finn 2018 [29] | 2018 | Len | 478 | 63 | 405/73 | 304/174/0 | 0/104/374/0 | NR | 135 | NR | 334 |
| | | Sor | 476 | 62 | 401/75 | 301/175/0 | 0/92/384/0 | NR | 141 | NR | 326 |
| Kudo Cheng 2018 [30] | 2018 | Ora | 444 | 67 | 363/81 | 401/43/0 | 158/209/74/0* | 170 | 0 | 0 | 444 |
| | | Pla | 444 | 66 | 364/80 | 406/38/0 | 135/229/72/0* | 202 | 0 | 0 | 444 |
| Meyer 2017 [31] | 2017 | Sor | 157 | 65 | 139/18 | 98/58/0* | NR | 15 | 157 | 0 | 0 |
| | | Pla | 156 | 68 | 138/18 | 97/58/0* | NR | 14 | 156 | 0 | 0 |
| Lee 2017 [32] | 2017 | Sor | 36 | 60 | 30/6 | NR | 9/27/0/0 | NR | 0 | 0 | 36 |
| | | Pla | 36 | 62 | 32 | NR | 15/21/0/0 | NR | 0 | 0 | 36 |
| Lencioni 2016 [33] | 2016 | Sor | 154 | 64.5 | 135/19 | 154/0/0 | 0/154/0/0 | 60 | 78 | NR | 59 |
| | | Pla | 153 | 63 | 126.27 | 153/0/0 | 0/153/0/0 | 51 | 79 | NR | 57 |
| Koeberle 2016 [34] | 2016 | Eve + Sor | 59 | 66 | 48/18 | 35/24/0 | 0/15/44/0 | 10 | 59 | 0 | 0 |
| | | Sor | 46 | 65 | 40/15 | 33/13/0 | 0/14/32/0 | 8 | 46 | 0 | 0 |
| Cheng 2016 [35] | 2016 | Dov | 82 | 56 | 73/9 | 52/30/0 | 0/2/80/0 | NR | 0 | 0 | 82 |
| | | Sor | 83 | 56 | 67/16 | 53/29/0* | 0/2/81/0 | NR | 0 | 0 | 83 |
| Zhu 2015 [36] | 2015 | Erl + Sor | 362 | 60.5 | 295/67 | 222/140/0 | 0/60/302/0 | 122 | 186 | NR | 88 |
| | | Sor | 358 | 60 | 286/72 | 216/142/0 | 0/48/310/0 | 133 | 183 | NR | 90 |
| Cheng 2015 [37] | 2015 | Tig 6mg + Sor | 54 | 62.5 | 45/9 | 31/23/0 | NR | 45 | NR | NR | 53 |
| | | Tig 2mg + Sor | 53 | 63 | 45/8 | 32/21/0 | NR | 33 | NR | NR | 52 |
| | | Sor | 55 | 66 | 44/11 | 30/25/0* | NR | 25 | NR | NR | 54 |
| Cainap 2015 [38] | 2015 | Lin | 514 | 59 | 444/70 | 323/191/0 | 0/81/433/0 | 251 | NR | NR | 339 |
| | | Sor | 521 | 60 | 436/85 | 344/176/0 | 0/102/418/0 | 257 | NR | NR | 350 |
| Kudo 2014 [39] | 2014 | Bri | 249 | 57 | 206/43 | 201/48/0 | 65/129/54/1 | 158 | 22 | NR | 216 |
| | | Pla | 253 | 59 | 216/37 | 203/50/0 | 57/150/44/2 | 168 | 23 | NR | 218 |
| Johnson 2013 [40] | 2013 | Bri | 577 | 61 | 483/94 | 361/216 | 37/95/444/0 | 254 | 134 | NR | 346 |
| | | Sor | 578 | 60 | 484/94 | 352/226 | 30/97/449/0 | 258 | 135 | NR | 372 |
| Inaba 2013 [41] | 2013 | Ora | 50 | NR | 39/11 | 45/5/0 | 21/24/5/0 | 2 | NR | NR | 50 |
| | | Pla | 51 | NR | 43/8 | 49/2/0 | 22/27/2/0 | 4 | NR | NR | 51 |
| Cheng 2013 [42] | 2013 | Sun | 530 | 59 | 436/94 | 278/248/0* | 0/67/462/0 | 290 | 111 | 6 | 411 |
| | | Sor | 544 | 59 | 459/85 | 288/254/0* | 0/89/454/0 | 288 | 112 | 10 | 418 |
| Hsu 2012 [43] | 2012 | Van 300mg | 19 | 54 | 18/1 | NR | 0/4/15/0 | 14 | NR | NR | 19 |
| | | Van 100mg | 25 | 61 | 17/8 | NR | 0/4/21/0 | 16 | NR | NR | 25 |
| | | Pla | 23 | 56 | 20/3 | NR | 0/5/18/0 | 17 | NR | NR | 23 |
| Kudo 2011 [44] | 2011 | Sor | 229 | 69 | 174/55 | 201/28/0 | NR | 47 | NR | NR | 229 |
| | | Pla | 229 | 70 | 168/61 | 202/27/0 | NR | 52 | NR | NR | 229 |
| Chen 2009 [45] | 2009 | Sor | 150 | 51 | 127/23 | 38/104/8 | 0/0/143/0* | 106 | NR | NR | 150 |
| | | Pla | 76 | 52 | 66/10 | 21/51/4 | 0/0/73/0* | 59 | NR | NR | 76 |
| Llovet 2008 [6] | 2008 | Sor | 299 | NR | 260/39 | 161/114/24 | 0/54/244/0 | 56* | NR | NR | NR |
| | | Pla | 303 | NR | 264/39 | 164/117/22 | 0/51/252/0 | 55* | NR | NR | NR |

* Data were not available for all patients; NR: Not report.

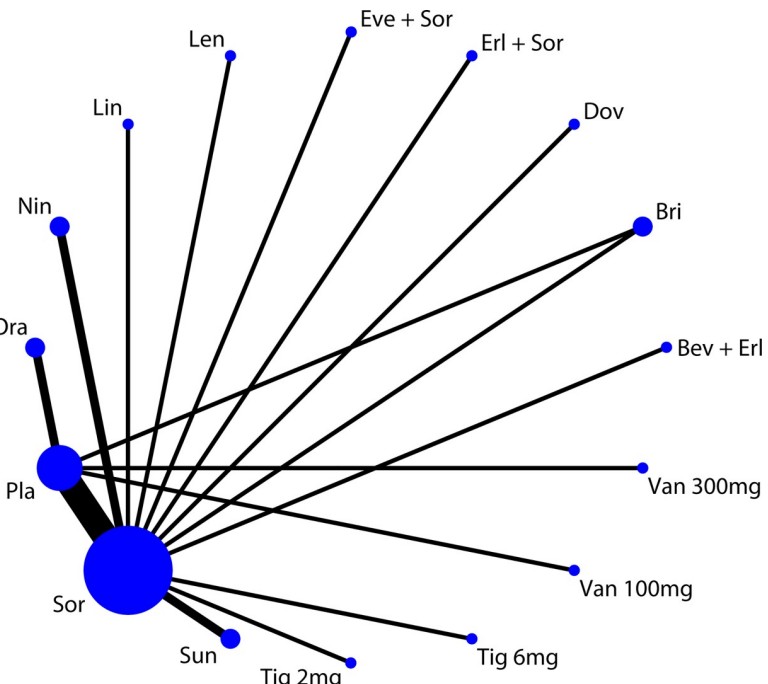

**Fig 2. Network diagram of all studies.**

95%CrI: 0.03–0.43), as shown in S2 Table. The NMA synthesis showed that four drugs (brivanib, lenvatinib, linifanib and sorafenib) achieved a significant benefit on TTP over placebo (HR range, 0.45–0.72). According to SUCRA, three highest ranking drugs were lenvatinib (0.94), linifanib (0.84) and brivanib (0.67), which were in red in Table 2.

## Progression-free survival

Eight RCTs [25, 26, 28, 29, 31, 38, 41, 43] reporting information on PFS were included for meta-analysis. Direct meta-analyses (S4 Fig) confirmed a significant improvement on PFS compared to Lenvatinib (HR: 0.66; 95%CI: 0.56–0.77) and Linifanib (HR: 0.81; 95%CI: 0.69–0.95) over sorafenib. A star-shaped network diagram formed by PFS is provided in S5 Fig. For no closed loop, node-splitting test of studies for PFS was not applicable. The NMA heterogeneity was low (σ = 0.18; 95%CrI: 0.01–0.43), as shown in S2 Table. The NMA synthesis showed that there was no significant difference on PFS among drugs. According to SUCRA, three highest ranking drugs were lenvatinib (0.77), vandetanib (0.77) and orantinib (0.68) which were in red in Table 3.

## Overall survival

Twenty RCTs [6, 25, 27–32, 34–45] reporting information on OS were included for meta-analysis. Direct meta-analyses (S6 Fig) confirmed a significant improvement on OS compared to sorafenib (HR: 0.72; 95%CI: 0.54–0.94) and Vandetanib 100 mg (HR: 0.44; 95%CI: 0.22–0.87) over placebo. A connected network diagram formed by OS is provided in S7 Fig. According to the node-splitting analysis, there was not any significant inconsistency between the direct and indirect comparisons (Pla vs. Bri, P = 0.62; Sor vs. Bri, P = 0.61; Sor vs. Pla, P = 0.62), as shown in S8 Fig. The NMA heterogeneity was low (σ = 0.15; 95%CrI: 0.01–0.49), as shown in S2 Table. The NMA synthesis showed that two treatments (Vandetanib 100 mg and sorafenib)

**Table 2. Network meta-analyses for TTP (Findings are expressed as HR (95% CrI), use of random-effect model).**

| SUCRA | Drugs | Bri | Erl+Sor | Eve+Sor | Len | Lin | Nin | Ora | Pla | Sor | Sun | Tig 2mg + Sor | Tig 6mg + Sor |
|---|---|---|---|---|---|---|---|---|---|---|---|---|---|
| 0.67 | Bri | Bri | 1.21 (0.67, 2.21) | 1.05 (0.54, 2.08) | 0.67 (0.37, 1.21) | 0.80 (0.45, 1.47) | 1.45 (0.80, 2.66) | 1.27 (0.71, 2.33) | 1.48 (1.04, 2.14) | 1.06 (0.75, 1.53) | 1.33 (0.84, 2.33) | 1.19 (0.62, 2.28) | 1.22 (0.65, 2.32) |
| 0.43 | Erl+Sor | 0.83 (0.45, 1.50) | Erl+Sor | 0.87 (0.41, 1.82) | 0.55 (0.28, 1.07) | 0.66 (0.34, 1.30) | 1.20 (0.61, 2.38) | 1.05 (0.52, 2.11) | 1.22 (0.73, 2.06) | 0.88 (0.54, 1.41) | 1.10 (0.63, 2.12) | 0.99 (0.47, 2.02) | 1.01 (0.49, 2.06) |
| 0.58 | Eve+Sor | 0.95 (0.48, 1.84) | 1.15 (0.55, 2.43) | Eve+Sor | 0.64 (0.30, 1.32) | 0.77 (0.37, 1.60) | 1.38 (0.65, 2.93) | 1.21 (0.56, 2.59) | 1.41 (0.76, 2.59) | 1.01 (0.57, 1.78) | 1.27 (0.66, 2.56) | 1.13 (0.51, 2.52) | 1.16 (0.53, 2.55) |
| 0.94 | Len | 1.50 (0.83, 2.67) | 1.82 (0.93, 3.53) | 1.57 (0.76, 3.29) | Len | 1.21 (0.62, 2.38) | 2.17 (1.12, 4.27) | 1.91 (0.95, 3.81) | 2.22 (1.33, 3.71) | 1.59 (1.00, 2.53) | 1.99 (1.15, 3.78) | 1.79 (0.86, 3.64) | 1.84 (0.90, 3.73) |
| 0.84 | Lin | 1.24 (0.68, 2.22) | 1.50 (0.77, 2.95) | 1.30 (0.63, 2.74) | 0.83 (0.42, 1.61) | Lin | 1.80 (0.92, 3.53) | 1.58 (0.78, 3.18) | 1.84 (1.09, 3.11) | 1.32 (0.82, 2.12) | 1.65 (0.95, 3.15) | 1.48 (0.71, 3.04) | 1.52 (0.74, 3.14) |
| 0.23 | Nin | 0.69 (0.38, 1.24) | 0.83 (0.42, 1.63) | 0.72 (0.34, 1.53) | 0.46 (0.23, 0.89) | 0.56 (0.28, 1.09) | Nin | 0.88 (0.43, 1.77) | 1.02 (0.60, 1.73) | 0.73 (0.45, 1.19) | 0.92 (0.52, 1.74) | 0.82 (0.39, 1.71) | 0.84 (0.41, 1.74) |
| 0.37 | Ora | 0.79 (0.43, 1.40) | 0.95 (0.48, 1.93) | 0.82 (0.39, 1.78) | 0.52 (0.26, 1.05) | 0.63 (0.31, 1.27) | 1.14 (0.57, 2.32) | Ora | 1.16 (0.73, 1.84) | 0.83 (0.50, 1.39) | 1.05 (0.58, 2.05) | 0.94 (0.44, 1.97) | 0.96 (0.46, 2.01) |
| 0.16 | Pla | 0.68 (0.47, 0.96) | 0.82 (0.48, 1.37) | 0.71 (0.39, 1.32) | 0.45 (0.27, 0.75) | 0.54 (0.32, 0.92) | 0.98 (0.58, 1.66) | 0.86 (0.54, 1.37) | Pla | 0.72 (0.58, 0.89) | 0.90 (0.61, 1.43) | 0.80 (0.44, 1.45) | 0.83 (0.47, 1.47) |
| 0.61 | Sor | 0.94 (0.66, 1.33) | 1.14 (0.71, 1.84) | 0.99 (0.56, 1.76) | 0.63 (0.39, 1.00) | 0.76 (0.47, 1.22) | 1.37 (0.84, 2.23) | 1.20 (0.72, 2.00) | 1.40 (1.13, 1.74) | Sor | 1.26 (0.91, 1.90) | 1.12 (0.64, 1.95) | 1.15 (0.68, 1.98) |
| 0.29 | Sun | 0.75 (0.43, 1.19) | 0.91 (0.47, 1.59) | 0.79 (0.39, 1.51) | 0.50 (0.26, 0.87) | 0.61 (0.32, 1.05) | 1.09 (0.58, 1.93) | 0.96 (0.49, 1.71) | 1.11 (0.70, 1.63) | 0.80 (0.53, 1.10) | Sun | 0.89 (0.44, 1.66) | 0.92 (0.47, 1.69) |
| 0.45 | Tig 2mg + Sor | 0.84 (0.44, 1.62) | 1.02 (0.50, 2.13) | 0.88 (0.40, 1.98) | 0.56 (0.28, 1.16) | 0.68 (0.33, 1.40) | 1.22 (0.59, 2.57) | 1.07 (0.51, 2.29) | 1.24 (0.69, 2.27) | 0.89 (0.51, 1.55) | 1.12 (0.60, 2.26) | Tig 2mg + Sor | 1.03 (0.66, 1.61) |
| 0.41 | Tig 6mg + Sor | 0.82 (0.43, 1.54) | 0.99 (0.49, 2.03) | 0.86 (0.39, 1.87) | 0.54 (0.27, 1.11) | 0.66 (0.32, 1.34) | 1.18 (0.58, 2.46) | 1.04 (0.50, 2.18) | 1.21 (0.68, 2.15) | 0.87 (0.51, 1.47) | 1.09 (0.59, 2.15) | 0.97 (0.62, 1.52) | Tig 6mg + Sor |

The values in red shading were the highest three SUCRAs. The values in green shading were statistically significant. The texts in yellow shading were targeted drugs.

achieved a significant benefit on OS over placebo (HR range, 0.44–0.73). According to SUCRA, three highest ranking interventions were tigatuzumab 6mg (0.73), vandetanib 100mg (0.92) and vandetanib 300mg (0.70), which were in red in Table 4.

## Objective response rates

Thirteen RCTs [6, 26, 28, 29, 31–33, 35, 36, 38–40, 45] reporting information on ORR were included for meta-analysis. Direct meta-analyses (S9 Fig) confirmed that ORR was better in case of lenvatinib (HR: 3.11; 95%CI: 2.14–4.52) or linifanib (HR: 1.72; 95%CI: 1.09–2.72) than sorafenib, and ORR was bad in case of brivanib (HR: 0.21; 95%CI: 0.14–0.31) or sunitinib (HR: 0.42; 95%CI: 0.19–0.93) than sorafenib. A connected network diagram formed by ORR is provided in S10 Fig. According to the node-splitting analysis, there was not any significant inconsistency between the direct and indirect comparisons (Pla vs. Bri, P = 0.13; Sor vs. Bri, P = 0.13; Sor vs. Pla, P = 0.13), as shown in S11 Fig. The NMA heterogeneity was low (σ = 0.72;

**Table 3. Network meta-analyses for PFS (Findings are expressed as HR (95% CrI), use of random-effect model).**

| SUCRA | Drugs | Len | Lin | Nin | Ora | Pla | Sor | Sun | Van 100mg | Van 300mg |
|---|---|---|---|---|---|---|---|---|---|---|
| 0.77 | Len | Len | 1.23 (0.58, 2.59) | 1.70 (0.84, 3.44) | 1.07 (0.39, 2.94) | 1.53 (0.70, 3.26) | 1.51 (0.90, 2.55) | 2.16 (0.95, 4.89) | 0.97 (0.35, 2.66) | 1.08 (0.40, 2.91) |
| 0.58 | Lin | 0.81 (0.39, 1.72) | Lin | 1.38 (0.67, 2.83) | 0.87 (0.32, 2.40) | 1.24 (0.58, 2.66) | 1.23 (0.72, 2.09) | 1.76 (0.77, 3.97) | 0.79 (0.29, 2.17) | 0.88 (0.32, 2.37) |
| 0.26 | Nin | 0.59 (0.29, 1.20) | 0.72 (0.35, 1.49) | Nin | 0.63 (0.24, 1.68) | 0.90 (0.43, 1.88) | 0.89 (0.55, 1.46) | 1.27 (0.57, 2.84) | 0.57 (0.21, 1.56) | 0.64 (0.24, 1.68) |
| 0.68 | Ora | 0.93 (0.34, 2.53) | 1.15 (0.42, 3.11) | 1.58 (0.60, 4.24) | Ora | 1.43 (0.75, 2.74) | 1.41 (0.60, 3.33) | 2.01 (0.70, 5.79) | 0.91 (0.36, 2.33) | 1.01 (0.41, 2.53) |
| 0.32 | Pla | 0.65 (0.31, 1.42) | 0.81 (0.38, 1.73) | 1.11 (0.53, 2.35) | 0.70 (0.37, 1.34) | Pla | 0.99 (0.57, 1.75) | 1.42 (0.61, 3.28) | 0.64 (0.32, 1.26) | 0.71 (0.37, 1.35) |
| 0.35 | Sor | 0.66 (0.39, 1.11) | 0.81 (0.48, 1.38) | 1.12 (0.68, 1.83) | 0.71 (0.30, 1.67) | 1.01 (0.57, 1.76) | Sor | 1.43 (0.75, 2.69) | 0.64 (0.27, 1.53) | 0.72 (0.30, 1.66) |
| 0.11 | Sun | 0.46 (0.20, 1.06) | 0.57 (0.25, 1.30) | 0.79 (0.35, 1.74) | 0.50 (0.17, 1.43) | 0.71 (0.30, 1.65) | 0.70 (0.37, 1.33) | Sun | 0.45 (0.15, 1.33) | 0.50 (0.17, 1.44) |
| 0.77 | Van 100mg | 1.03 (0.38, 2.83) | 1.26 (0.46, 3.46) | 1.75 (0.64, 4.73) | 1.10 (0.43, 2.81) | 1.57 (0.80, 3.08) | 1.56 (0.65, 3.72) | 2.23 (0.75, 6.54) | Van 100mg | 1.11 (0.65, 1.89) |
| 0.66 | Van 300mg | 0.93 (0.34, 2.50) | 1.14 (0.42, 3.09) | 1.57 (0.59, 4.17) | 0.99 (0.40, 2.47) | 1.41 (0.74, 2.69) | 1.40 (0.60, 3.29) | 2.00 (0.69, 5.79) | 0.90 (0.53, 1.54) | Van 300mg |

The values in red shading were the highest three SUCRAs. The values in green shading were statistically significant. The texts in yellow shading were targeted drugs.

95%CrI: 0.31–1.45), as shown in S2 Table. The NMA synthesis showed that there was no significant difference on ORR among drugs. According to SUCRA, three highest ranking interventions were lenvatinib (0.88), erlotinib plus sorafenib (0.73) and linifanib (0.73) which were in red in Table 5.

## The proportion of Grade 3–5 adverse events

Eleven RCTs [6, 25, 28, 34–36, 38–40, 43, 45] reporting information on G3-5AE were included for meta-analysis. Direct meta-analyses (S12 Fig) confirmed that brivanib (HR: 0.14; 95%CI: 0.10–0.21) and nintedanib (HR: 0.23; 95%CI: 0.10–0.52) than sorafenib. A connected network diagram formed by G3-5AE was provided in S13 Fig. According to the node-splitting analysis, there was not any significant inconsistency between the direct and indirect comparisons (Pla vs. Bri, $P$ = 0.25; Sor vs. Bri, $P$ = 0.25; Sor vs. Pla, $P$ = 0.25), as shown in S14 Fig. The NMA heterogeneity was low (σ = 0.99; 95%CrI: 0.42–1.92), as shown in S2 Table. The NMA synthesis showed that there was no significant difference on G3-5AE among drugs. According to SUCRA, three highest ranking interventions were vandetanib (vandetanib 100 mg twice daily [0.89]; vandetanib 300 mg twice daily [0.82]) and nintedanib (0.67), which were in red in Table 6.

## Cluster rank analysis

According to the meta-analysis performed above, ten interventions (Bri, Dov, Erl + Sor, Eve + Sor, Lin, Nin, Pla, Sor, Van 100mg and Van 300mg) compared to each other head-to-head on both OS and G3-5AE. According to cluster rank analysis, Van 100mg was the drug with both more effective (OS) and more secure (G3-5AE) compared to Sor followed by Nin (Fig 3).

## Consistency, heterogeneity and quality of evidence

The detection of inconsistency in evidence networks was conducted by evaluating the agreement between the consistency and inconsistency (unrelated mean effects) models (S3 Table).

**Table 4. Network meta-analyses for OS (Findings are expressed as HR (95% CrI), use of random-effect model).**

| SUCRA | Drugs | Bev +Erl | Bri | Dov | Erl +Sor | Eve +Sor | Len | Lin | Nin | Ora | Pla | Sor | Sun | Tig 2mg + Sor | Tig 6mg + Sor | Van 100mg | Van 300mg |
|---|---|---|---|---|---|---|---|---|---|---|---|---|---|---|---|---|---|
| 0.62 | Bev +Erl | Bev +Erl | 1.21 (0.60, 2.53) | 1.38 (0.59, 3.15) | 1.01 (0.47, 2.18) | 1.20 (0.51, 2.81) | 1.00 (0.46, 2.17) | 1.14 (0.52, 2.48) | 0.99 (0.45, 2.14) | 1.59 (0.75, 3.40) | 1.48 (0.76, 2.92) | 1.08 (0.57, 2.03) | 1.41 (0.64, 3.01) | 1.35 (0.58, 3.06) | 0.91 (0.39, 2.09) | 0.65 (0.26, 1.68) | 0.89 (0.35, 2.31) |
| 0.41 | Bri | 0.82 (0.39, 1.67) | Bri | 1.14 (0.58, 2.12) | 0.83 (0.45, 1.48) | 0.99 (0.49, 1.90) | 0.82 (0.45, 1.46) | 0.94 (0.51, 1.66) | 0.81 (0.45, 1.43) | 1.32 (0.78, 2.16) | 1.23 (0.83, 1.75) | 0.90 (0.61, 1.25) | 1.16 (0.62, 2.03) | 1.11 (0.56, 2.10) | 0.75 (0.38, 1.42) | 0.54 (0.25, 1.12) | 0.73 (0.34, 1.54) |
| 0.29 | Dov | 0.73 (0.32, 1.68) | 0.88 (0.47, 1.73) | Dov | 0.73 (0.36, 1.52) | 0.87 (0.39, 1.92) | 0.72 (0.35, 1.48) | 0.83 (0.40, 1.70) | 0.72 (0.35, 1.47) | 1.16 (0.58, 2.33) | 1.08 (0.60, 1.98) | 0.79 (0.46, 1.37) | 1.02 (0.49, 2.09) | 0.98 (0.45, 2.11) | 0.66 (0.31, 1.44) | 0.47 (0.20, 1.16) | 0.65 (0.27, 1.58) |
| 0.65 | Erl +Sor | 0.99 (0.46, 2.15) | 1.20 (0.68, 2.23) | 1.37 (0.66, 2.81) | Erl +Sor | 1.19 (0.57, 2.46) | 0.99 (0.51, 1.92) | 1.13 (0.58, 2.20) | 0.98 (0.51, 1.90) | 1.59 (0.85, 2.99) | 1.47 (0.87, 2.52) | 1.08 (0.67, 1.73) | 1.40 (0.71, 2.69) | 1.33 (0.64, 2.72) | 0.91 (0.44, 1.85) | 0.65 (0.28, 1.50) | 0.88 (0.38, 2.05) |
| 0.44 | Eve +Sor | 0.84 (0.36, 1.95) | 1.01 (0.53, 2.02) | 1.15 (0.52, 2.53) | 0.84 (0.41, 1.75) | Eve +Sor | 0.83 (0.40, 1.74) | 0.96 (0.46, 1.98) | 0.82 (0.40, 1.71) | 1.33 (0.65, 2.74) | 1.24 (0.67, 2.32) | 0.91 (0.51, 1.61) | 1.17 (0.57, 2.44) | 1.12 (0.51, 2.47) | 0.76 (0.34, 1.69) | 0.54 (0.22, 1.36) | 0.74 (0.31, 1.84) |
| 0.66 | Len | 1.00 (0.46, 2.18) | 1.21 (0.69, 2.22) | 1.38 (0.68, 2.85) | 1.01 (0.52, 1.97) | 1.20 (0.58, 2.52) | Len | 1.14 (0.60, 2.21) | 0.99 (0.52, 1.88) | 1.60 (0.86, 3.02) | 1.49 (0.89, 2.53) | 1.09 (0.68, 1.72) | 1.41 (0.73, 2.70) | 1.35 (0.66, 2.75) | 0.91 (0.45, 1.89) | 0.65 (0.29, 1.50) | 0.89 (0.39, 2.05) |
| 0.49 | Lin | 0.88 (0.40, 1.91) | 1.06 (0.60, 1.97) | 1.21 (0.59, 2.47) | 0.88 (0.45, 1.71) | 1.05 (0.51, 2.19) | 0.87 (0.45, 1.68) | Lin | 0.87 (0.45, 1.64) | 1.40 (0.76, 2.65) | 1.30 (0.78, 2.22) | 0.95 (0.60, 1.52) | 1.23 (0.63, 2.36) | 1.18 (0.57, 2.39) | 0.80 (0.39, 1.61) | 0.57 (0.25, 1.32) | 0.78 (0.34, 1.80) |
| 0.65 | Nin | 1.01 (0.47, 2.21) | 1.23 (0.70, 2.24) | 1.39 (0.68, 2.84) | 1.02 (0.53, 1.96) | 1.21 (0.58, 2.52) | 1.01 (0.53, 1.93) | 1.15 (0.61, 2.23) | Nin | 1.61 (0.88, 3.03) | 1.50 (0.90, 2.52) | 1.10 (0.69, 1.74) | 1.42 (0.75, 2.71) | 1.36 (0.67, 2.75) | 0.92 (0.45, 1.89) | 0.66 (0.29, 1.52) | 0.90 (0.40, 2.09) |
| 0.13 | Ora | 0.63 (0.29, 1.32) | 0.76 (0.46, 1.28) | 0.86 (0.43, 1.72) | 0.63 (0.33, 1.18) | 0.75 (0.36, 1.53) | 0.63 (0.33, 1.16) | 0.72 (0.38, 1.32) | 0.62 (0.33, 1.14) | Ora | 0.93 (0.65, 1.32) | 0.68 (0.44, 1.03) | 0.88 (0.46, 1.63) | 0.84 (0.42, 1.65) | 0.57 (0.29, 1.13) | 0.41 (0.20, 0.85) | 0.56 (0.27, 1.16) |
| 0.18 | Pla | 0.67 (0.34, 1.31) | 0.82 (0.57, 1.21) | 0.93 (0.51, 1.67) | 0.68 (0.40, 1.14) | 0.81 (0.43, 1.48) | 0.67 (0.40, 1.12) | 0.77 (0.45, 1.28) | 0.67 (0.40, 1.11) | 1.07 (0.76, 1.53) | Pla | 0.73 (0.57, 0.92) | 0.95 (0.56, 1.57) | 0.91 (0.50, 1.62) | 0.61 (0.34, 1.10) | 0.44 (0.23, 0.84) | 0.60 (0.31, 1.16) |
| 0.57 | Sor | 0.92 (0.49, 1.74) | 1.11 (0.80, 1.64) | 1.27 (0.73, 2.20) | 0.93 (0.58, 1.49) | 1.10 (0.62, 1.95) | 0.92 (0.58, 1.47) | 1.05 (0.66, 1.68) | 0.91 (0.58, 1.44) | 1.47 (0.97, 2.26) | 1.37 (1.09, 1.75) | Sor | 1.30 (0.81, 2.05) | 1.24 (0.72, 2.14) | 0.84 (0.49, 1.45) | 0.60 (0.30, 1.20) | 0.82 (0.41, 1.67) |
| 0.25 | Sun | 0.71 (0.33, 1.55) | 0.86 (0.49, 1.61) | 0.98 (0.48, 2.02) | 0.72 (0.37, 1.40) | 0.85 (0.41, 1.76) | 0.71 (0.37, 1.38) | 0.81 (0.42, 1.58) | 0.70 (0.37, 1.33) | 1.13 (0.61, 2.16) | 1.05 (0.64, 1.79) | 0.77 (0.49, 1.23) | Sun | 0.95 (0.47, 1.95) | 0.65 (0.32, 1.33) | 0.46 (0.21, 1.07) | 0.63 (0.28, 1.47) |
| 0.31 | Tig 2mg + Sor | 0.74 (0.33, 1.73) | 0.90 (0.48, 1.77) | 1.02 (0.47, 2.20) | 0.75 (0.37, 1.56) | 0.89 (0.40, 1.97) | 0.74 (0.36, 1.52) | 0.85 (0.42, 1.74) | 0.74 (0.36, 1.50) | 1.19 (0.60, 2.39) | 1.10 (0.62, 2.01) | 0.81 (0.47, 1.39) | 1.05 (0.51, 2.13) | Tig 2mg + Sor | 0.68 (0.43, 1.07) | 0.48 (0.20, 1.17) | 0.66 (0.28, 1.63) |
| 0.73 | Tig 6mg + Sor | 1.10 (0.48, 2.55) | 1.33 (0.70, 2.62) | 1.51 (0.70, 3.27) | 1.10 (0.54, 2.27) | 1.31 (0.59, 2.92) | 1.09 (0.53, 2.24) | 1.25 (0.62, 2.54) | 1.08 (0.53, 2.21) | 1.75 (0.89, 3.49) | 1.63 (0.91, 2.96) | 1.19 (0.69, 2.05) | 1.54 (0.75, 3.11) | 1.47 (0.93, 2.32) | Tig 6mg + Sor | 0.72 (0.30, 1.72) | 0.97 (0.41, 2.38) |
| 0.92 | Van 100mg | 1.53 (0.59, 3.90) | 1.86 (0.89, 3.92) | 2.11 (0.86, 5.04) | 1.54 (0.67, 3.51) | 1.84 (0.74, 4.48) | 1.53 (0.67, 3.46) | 1.75 (0.76, 3.94) | 1.51 (0.66, 3.43) | 2.45 (1.18, 5.05) | 2.27 (1.19, 4.33) | 1.66 (0.83, 3.29) | 2.15 (0.93, 4.88) | 2.06 (0.86, 4.91) | 1.39 (0.58, 3.33) | Van 100mg | 1.37 (0.85, 2.20) |
| 0.70 | Van 300mg | 1.12 (0.43, 2.89) | 1.36 (0.65, 2.91) | 1.55 (0.63, 3.71) | 1.13 (0.49, 2.60) | 1.35 (0.54, 3.27) | 1.12 (0.49, 2.54) | 1.29 (0.55, 2.91) | 1.11 (0.48, 2.52) | 1.79 (0.86, 3.74) | 1.67 (0.86, 3.20) | 1.22 (0.60, 2.43) | 1.58 (0.68, 3.59) | 1.51 (0.61, 3.60) | 1.03 (0.42, 2.44) | 0.73 (0.46, 1.17) | Van 300mg |

The values in red shading were the highest three SUCRAs. The values in green shading were statistically significant. The texts in yellow shading were targeted drugs.

The results of comparisons in both consistency and inconsistency models were roughly consistent. The result showed a robust and homogeneous network of evidence. Additionally, the node-splitting approach also showed a good consistency between the direct and indirect

**Table 5. Network meta-analyses for ORR (Findings are expressed as OR (95% CrI), use of random-effect model).**

| SUCRA | Drugs | Bri | Dov | Erl+Sor | Len | Lin | Pla | Sor | Sun |
|---|---|---|---|---|---|---|---|---|---|
| 0.19 | Bri | Bri | 1.41 (0.13, 16.54) | 4.83 (0.53, 48.67) | 8.54 (0.97, 76.55) | 4.73 (0.54, 43.77) | 1.35 (0.36, 4.97) | 2.72 (0.76, 10.54) | 1.14 (0.12, 11.73) |
| 0.34 | Dov | 0.71 (0.06, 7.83) | Dov | 3.39 (0.22, 54.05) | 6.01 (0.41, 92.30) | 3.35 (0.22, 50.15) | 0.95 (0.10, 8.80) | 1.93 (0.25, 15.29) | 0.80 (0.05, 13.44) |
| 0.73 | Erl +Sor | 0.21 (0.02, 1.88) | 0.29 (0.02, 4.54) | Erl+Sor | 1.76 (0.14, 22.92) | 0.98 (0.08, 12.74) | 0.28 (0.03, 2.02) | 0.56 (0.09, 3.53) | 0.24 (0.02, 3.17) |
| 0.88 | Len | 0.12 (0.01, 1.03) | 0.17 (0.01, 2.46) | 0.57 (0.04, 7.15) | Len | 0.55 (0.05, 6.77) | 0.16 (0.02, 1.09) | 0.32 (0.06, 1.90) | 0.13 (0.01, 1.75) |
| 0.73 | Lin | 0.21 (0.02, 1.86) | 0.30 (0.02, 4.61) | 1.02 (0.08, 12.87) | 1.81 (0.15, 21.15) | Lin | 0.29 (0.04, 1.95) | 0.58 (0.10, 3.38) | 0.24 (0.02, 3.16) |
| 0.29 | Pla | 0.74 (0.20, 2.78) | 1.05 (0.11, 9.76) | 3.57 (0.50, 29.14) | 6.32 (0.92, 47.13) | 3.49 (0.51, 25.69) | Pla | 2.02 (0.88, 5.08) | 0.84 (0.11, 7.22) |
| 0.58 | Sor | 0.37 (0.09, 1.32) | 0.52 (0.07, 3.97) | 1.77 (0.28, 11.07) | 3.14 (0.53, 17.66) | 1.73 (0.30, 9.84) | 0.50 (0.20, 1.14) | Sor | 0.42 (0.06, 2.76) |
| 0.26 | Sun | 0.88 (0.09, 8.42) | 1.25 (0.07, 19.77) | 4.24 (0.32, 60.22) | 7.52 (0.57, 95.11) | 4.16 (0.32, 54.27) | 1.19 (0.14, 8.86) | 2.41 (0.36, 15.64) | Sun |

The values in red shading were the highest three SUCRAs. The texts in yellow shading were targeted drugs.

comparisons (S3, S8, S11 and S14 Figs). Though application of a fixed-effect model would provide similar numerical results with shorter credible intervals, random-effect model was more appropriate according to the residual deviance and DIC criteria (S2 Table). There was no obvious asymmetry at visual inspection of funnel plots to suggest publication bias as shown in S16 Fig. According to GRADE approach, the direct, indirect, and NMA Estimates for OS and G3-5AE were shown in S4 and S5 Tables. The quality of the most evidence was low.

**Table 6. Network meta-analyses for G3-5AE (Findings are expressed as OR (95% CrI), use of random-effect model).**

| SUCRA | Drugs | Bri | Dov | Erl+Sor | Eve+Sor | Lin | Nin | Pla | Sor | Van 100mg | Van 300mg |
|---|---|---|---|---|---|---|---|---|---|---|---|
| 0.62 | Bri | Bri | 5.72 (0.28, 123.97) | 5.35 (0.25, 115.35) | 5.37 (0.26, 111.72) | 7.44 (0.37, 154.93) | 0.83 (0.06, 11.06) | 0.60 (0.09, 3.66) | 3.98 (0.62, 25.71) | 0.19 (0.01, 4.27) | 0.29 (0.01, 6.58) |
| 0.25 | Dov | 0.17 (0.01, 3.57) | Dov | 0.94 (0.03, 27.07) | 0.93 (0.03, 26.50) | 1.30 (0.04, 36.79) | 0.14 (0.01, 2.98) | 0.10 (0.01, 1.75) | 0.69 (0.06, 7.67) | 0.03 (0.00, 1.55) | 0.05 (0.00, 2.29) |
| 0.26 | Erl+Sor | 0.19 (0.01, 3.97) | 1.07 (0.04, 32.27) | Erl+Sor | 1.00 (0.03, 28.79) | 1.39 (0.05, 38.78) | 0.15 (0.01, 3.09) | 0.11 (0.01, 1.91) | 0.74 (0.07, 8.14) | 0.04 (0.00, 1.63) | 0.05 (0.00, 2.52) |
| 0.26 | Eve+Sor | 0.19 (0.01, 3.77) | 1.08 (0.04, 33.43) | 1.00 (0.03, 29.28) | Eve+Sor | 1.38 (0.05, 40.13) | 0.15 (0.01, 3.10) | 0.11 (0.01, 1.87) | 0.74 (0.07, 7.98) | 0.04 (0.00, 1.63) | 0.05 (0.00, 2.48) |
| 0.19 | Lin | 0.13 (0.01, 2.73) | 0.77 (0.03, 23.24) | 0.72 (0.03, 21.74) | 0.73 (0.02, 20.36) | Lin | 0.11 (0.01, 2.25) | 0.08 (0.00, 1.31) | 0.53 (0.05, 5.87) | 0.03 (0.00, 1.13) | 0.04 (0.00, 1.70) |
| 0.67 | Nin | 1.21 (0.09, 16.40) | 6.95 (0.34, 155.71) | 6.50 (0.32, 131.89) | 6.51 (0.32, 129.54) | 8.94 (0.44, 183.46) | Nin | 0.72 (0.06, 8.01) | 4.82 (0.77, 31.28) | 0.24 (0.01, 7.85) | 0.35 (0.01, 12.15) |
| 0.74 | Pla | 1.67 (0.27, 11.07) | 9.62 (0.57, 190.57) | 8.99 (0.52, 159.81) | 8.98 (0.53, 157.59) | 12.41 (0.76, 220.52) | 1.39 (0.12, 15.75) | Pla | 6.63 (1.45, 33.65) | 0.32 (0.02, 4.32) | 0.49 (0.03, 6.67) |
| 0.31 | Sor | 0.25 (0.04, 1.61) | 1.44 (0.13, 16.96) | 1.35 (0.12, 14.62) | 1.36 (0.13, 13.90) | 1.87 (0.17, 20.16) | 0.21 (0.03, 1.29) | 0.15 (0.03, 0.69) | Sor | 0.05 (0.00, 0.95) | 0.07 (0.00, 1.44) |
| 0.89 | Van 100mg | 5.16 (0.23, 122.61) | 29.84 (0.64, 1511.71) | 28.01 (0.61, 1342.11) | 27.66 (0.61, 1326.10) | 38.28 (0.89, 1848.26) | 4.23 (0.13, 156.02) | 3.08 (0.23, 40.53) | 20.56 (1.05, 438.34) | Van 100mg | 1.48 (0.10, 21.26) |
| 0.82 | Van 300mg | 3.48 (0.15, 88.32) | 19.97 (0.44, 1062.10) | 18.75 (0.40, 923.34) | 18.75 (0.40, 949.56) | 25.76 (0.59, 1255.14) | 2.88 (0.08, 107.45) | 2.06 (0.15, 29.58) | 13.79 (0.70, 308.28) | 0.68 (0.05, 9.82) | Van 300mg |

The values in red shading were three highest SUCRA. The texts in yellow shading were targeted drugs.

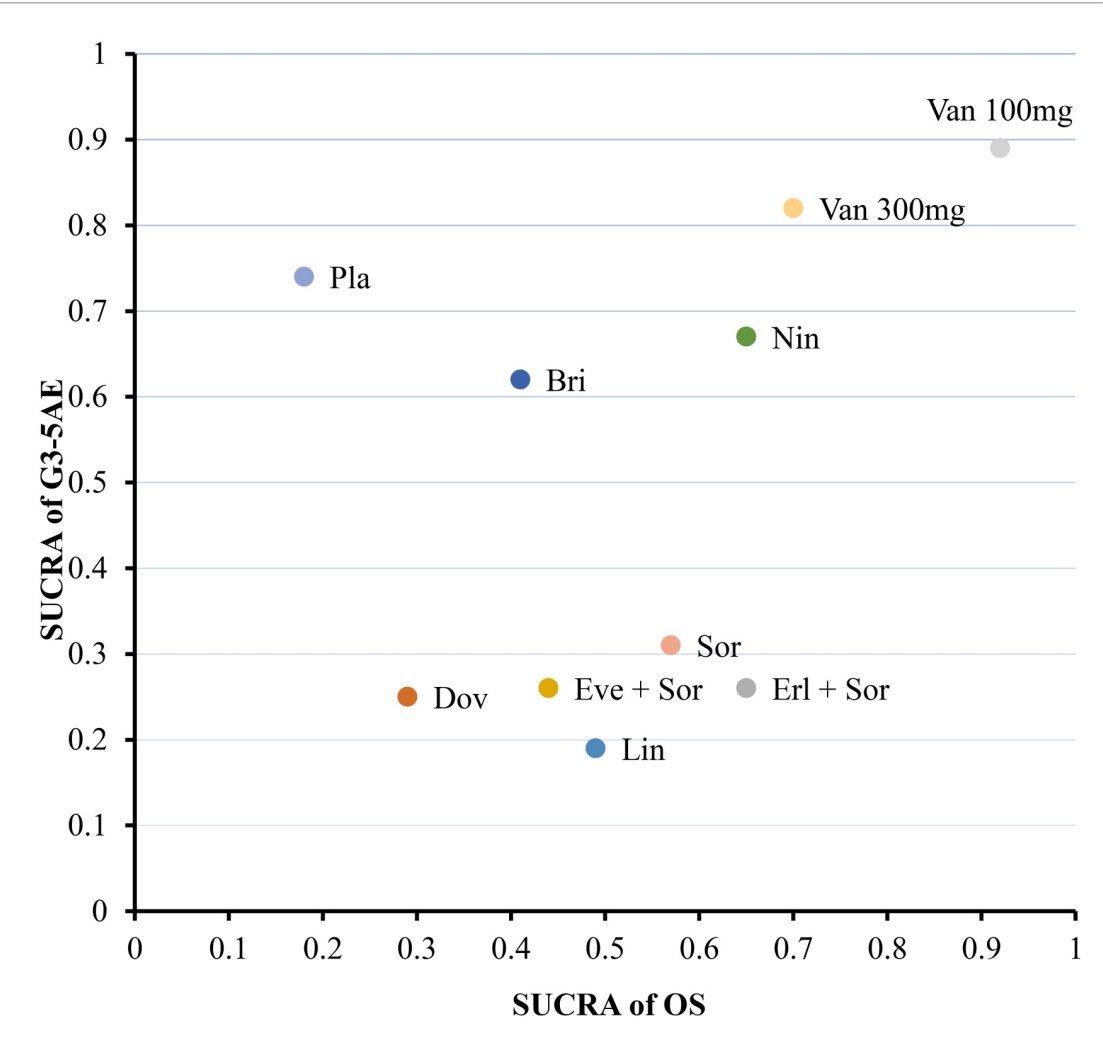

**Fig 3. Clustered ranking plot on OS and G3-5AE both expressed as SUCRAs.** The plot guides readers with respect to the trade-off between safety (G3-5AE) and effectiveness (OS) across the interventions. Interventions in the right upper corner tend to be more secure (higher SUCRA for G3-5AE) and more effective (higher SUCRA for OS) than those in the left lower corner of the plot.

## Discussion

The SHARP trial was the first study to demonstrate efficacy (HR = 0.69; 95% CI 0.55–0.87, for sorafenib vs placebo, on OS) of targeted therapy for patients with unresectable HCC [6]. Subsequently, an Asia-Pacific study also confirmed the same conclusion (HR = 0.68, 95% CI 0.50–0.93) [45]. Based on the results of the two trials, sorafenib, a multi-targeted TKI, became the standard systemic treatment, approved by the regulatory authorities around the world, for patients with advanced unresectable HCC [46]. However, the advantages of survival and the improvements of symptom or living quality in these two trials were modest. In order to find more effective targeted drugs, several clinical trials ensued. Disappointingly, most of the results were negative.

Several targeted drugs were compared with sorafenib directly in this review [25–29, 34–38, 40, 42]. For TTP, only Len (HR = 0.63, 95% CI 0.54–0.74) and Lin (HR = 0.76, 95% CrI 0.64–0.91) performed better than sorafenib while others comparisons showed no statistical

difference. For PFS, also Len (HR = 0.66, 95% CrI 0.56–0.77) and Lin (HR = 0.81, 95% CrI 0.69–0.95) performed better than sorafenib while others comparisons showed no statistical difference. For OS, no targeted drugs were superior to sorafenib while Sun performed worse than sorafenib with statistical difference. These direct comparison results are disappointing. Gratifyingly, a RCT verified that Van 100mg was superior in improving OS compared to placebo, although it didn't indicated that Van 100mg was better than sorafenib.

To see the results of different targeted drugs comparing to each other, we performed this Bayesian network analysis. In this meta-analysis, brivanib, lenvatinib and linifanib were superior in improving TTP compared to placebo. However, they showed non-superiority in terms of both PFS and OS compared with placebo. Sorafenib was superior in improving both TTP and OS, while Van 100mg was also superior in improving OS. Although Tig 6mg + Sor, Van 300mg and Van 100mg were the three highest ranking interventions, they showed non-superiority in terms of OS compared with sorafenib. For ORR and G3-5AE, there was no significant difference across all targeted drugs. In general, sorafenib appeared to remain superior in the present analysis.

There are some potential reasons for failure to meet the primary endpoints of prolonging OS in HCC trials. First, the inclusion criteria of clinical trials are mainly based on Child-Pugh scores and BCLC stages. However, this screening method couldn't eliminate the histologic heterogeneity in HCC. Therefore, several biomarkers (e.g., c-MET, RAS and FGF19) were recently used as bases for screening [47, 48]. Lack of predictive biomarkers was also one of the reasons for the failure. Second, by analyzing the target of included drugs, most of the drugs were anti-angiogenic multikinase inhibitors sharing some common pathways [49]. For these trials, there must be only marginal differences relative to sorafenib. To avoid similar targets, several trails tested a new drug in combination with sorafenib vs sorafenib alone, for instance, erlotinib targeting epidermal growth factor receptor, and everolimus targeting mammalian target of rapamycin. However, none of these combinations were superior in improving OS compared to sorafenib. Therefore, there still must be some other reasons for failure in HCC trials. Third, the end point OS is affected by advanced cirrhosis since advanced HCC is often accompanied by severe cirrhosis. The differences in curative effects among targeted drugs may not enough to cause major improvements in survival. To some extent, TTP may more suitable as an endpoint in advanced hepatocellular carcinoma treated with molecular targeted therapy [50]. Fourth, liver cirrhosis is frequently associated with hypohepatia. Due to the insufficiency of liver's synthesis and metabolism function, the expected drug effect may not be exerted. Meanwhile, the side effects of drugs often lead to treatment interruption.

According to the cluster rank analysis, Van 100mg, Van 300mg and Nintedanib were more effective and more secure compared to Sorafenib, although the advantages were not statistically significant. Although vandetanib has limited clinical activity and was not warranted to be further developed as first-line therapy for advanced HCC [43], the correlational research of vandetanib in HCC had not stopped. Vandetanib-eluting radiopaque beads for locoregional treatment of HCC were under development [51]. Recent studies showed that nintedanib might have similar efficacy comparing to sorafenib in patients with advanced HCC, but with a manageable safety profile [25].

As we know, this is the first network meta-analysis of all RCTs to evaluate the efficacy and safety of targeted drugs for the treatment of HCC patients. Several limitations should be taken into consideration. First, the distributions of BCLC stages in different studies were not in full accord. Patients with B or C stage often had worse prognosis than those with A stage. The BCLC criteria for the patients could have an impact on the overall survival. Fortunately, the vast majority of patients include in this analysis were in stage B or C. Second, cirrhosis is also an important correlation factor in survival. Third, some HRs [26] were obtained by calculating

the data extracted from the survival curves when they could not be acquired from the original article directly. Forth, both Response Evaluation Criteria in Solid Tumors (RECIST) v1.0, RECIST v1.1 and Modified RECIST (mRECIST) were used in the included studies. Both National Cancer Institute Common Terminology Criteria for Adverse Events, Version 3.0 and Version 4.0 were used in the included studies.

Our study also has several superiorities. First, we performed a comprehensive literature search to provide a summary of targeted therapies on HCC as detailed as possible. Second, in contrast to previous meta-analyses, the included studies were all RCTs that ensured the reliability of evidences. Third, we performed the cluster rank analysis considering both efficiency and safety in order to support clinical decision.

## Conclusion

Taken together, our network meta-analysis suggests that vandetanib, linifanib, lenvatinib and nintedanib potentially may be the best substitution of sorafenib against advanced HCC. For OS, Van (100 and 300mg), seem to be the best options with low and moderate quality of evidence, respectively. For G3-5AE, Van (100 and 300mg), seem to be the best interventions, with low and very low quality of evidence all of them. Further studies are necessary to explore the curative effect of certain subgroup in HCC patients, especially the subgroup classified as BCLC stage, Child-Pugh score and Hepatitis B infection status. For better survival, novel targeted treatment options for HCC are sorely needed.

## Supporting information

**S1 File. PRISMA 2009 flow diagram.**
(DOCX)

**S2 File. Detailed search strategy.**
(DOCX)

**S3 File. Targeted drug treatment programs.**
(DOCX)

**S1 Table. Risk of bias of included studies.**
(DOCX)

**S2 Table. Heterogeneity and model fit.**
(DOCX)

**S3 Table. Inconsistency analysis of treatment effects (random effects models—95% CrI).**
(DOCX)

**S4 Table. Direct, indirect, and NMA estimates for OS with the GRADE assessment.**
(DOCX)

**S5 Table. Direct, indirect, and NMA estimates for G3-5AE with the GRADE assessment.**
(DOCX)

**S1 Fig. Forest plot (random effects) of direct meta-analyses for TTP.**
(TIF)

**S2 Fig. Network diagram of studies for TTP.**
(TIF)

**S3 Fig. Node-splitting test of studies for TTP.**
(TIF)

**S4 Fig. Forest plot (random effects) of direct meta-analyses for PFS.**
(TIF)

**S5 Fig. Network diagram of studies for PFS.**
(TIF)

**S6 Fig. Forest plot (random effects) of direct meta-analyses for OS.**
(TIF)

**S7 Fig. Network diagram of studies for OS.**
(TIF)

**S8 Fig. Node-splitting test of studies for OS.**
(TIF)

**S9 Fig. Forest plot (random effects) of direct meta-analyses for ORR.**
(TIF)

**S10 Fig. Network diagram of studies for ORR.**
(TIF)

**S11 Fig. Node-splitting test of studies for ORR.**
(TIF)

**S12 Fig. Forest plot (random effects) of direct meta-analyses for G3-5AE.**
(TIF)

**S13 Fig. Network diagram of studies for G3-5AE.**
(TIF)

**S14 Fig. Node-splitting test of studies for G3-5AE.**
(TIF)

**S15 Fig.**
(TIF)

**S16 Fig. Comparison-adjusted funnel plots for all comparisons.**
(TIF)

## Author Contributions

**Conceptualization:** Xuezhong Xu.

**Data curation:** Wei Ding, Yulin Tan.

**Formal analysis:** Wenbo Xue.

**Investigation:** Wei Ding.

**Methodology:** Yulin Tan, Yan Qian, Wenbo Xue, Yibo Wang, Peng Jiang.

**Software:** Yan Qian, Peng Jiang.

**Validation:** Wei Ding, Xuezhong Xu.

**Writing – original draft:** Wei Ding.

**Writing – review & editing:** Xuezhong Xu.

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
