## [Decision Letter · Decision Letter 0]

26 Sep 2019

PONE-D-19-22935

First-line targeted therapies of advanced hepatocellular carcinoma: A Bayesian network analysis of randomized controlled trials

PLOS ONE

Dear Mr Xu,

Thank you for submitting your manuscript to PLOS ONE. After careful consideration, we feel that it has merit but does not fully meet PLOS ONE’s publication criteria as it currently stands. Therefore, we invite you to submit a revised version of the manuscript that addresses the points raised during the review process.

We would appreciate receiving your revised manuscript by Nov 10 2019 11:59PM. To enhance the reproducibility of your results, we recommend that if applicable you deposit your laboratory protocols in protocols.io, where a protocol can be assigned its own identifier (DOI) such that it can be cited independently in the future. For instructions see: http://journals.plos.org/plosone/s/submission-guidelines#loc-laboratory-protocols

We look forward to receiving your revised manuscript.

Kind regards,

Ivan D. Florez

Academic Editor

PLOS ONE

Journal Requirements:

1. Thank you for including your funding statement; "The funders had no role in study design, data collection and analysis, decision to publish, or preparation of the manuscript."

Please provide an amended Funding Statement that declares *all* the funding or sources of support received during this specific study (whether external or internal to your organization) as detailed online in our guide for authors at http://journals.plos.org/plosone/s/submit-now.  

Please state what role the funders took in the study.  If any authors received a salary from any of your funders, please state which authors and which funder. If the funders had no role, please state: "The funders had no role in study design, data collection and analysis, decision to publish, or preparation of the manuscript."

Additional Editor Comments (if provided):

Dear Authors,

Your manuscript has been evaluated by two reviewers who have raised a number of points that need to be addressed before considering it for publication. Please provide a response to each one fo the points described below. In particular, make emphasis on the following points which may be the most problematic, and are esssential for any NMA:

- Protocol: Reviewer 1 has raised a crucial point regarding the protocol for this nma. One criterion for any systematic review to be of high quality is the presence of a protocol before starting the review. You haven't provided evidence that that protocol existed. At least a register in a database for Systematic reviews should have existed (e.eg., PROSPERO database). Please, either: provide evidence that the protocol/registration existed and was available before stariting the project or explicitly state in the mauscript that the review was not registered in PROSPERO database and did not have an online protocll available, and provide a brief explanation.

- Inconsistency. Reviewer #1 has raised the point on the inconsistency tests. You should explicitly provide the description of the global and the local tests performed to define the inconsistency (per loop), and provide the results of all the tests: the global and each one of the local tests perfomed for each outcome, and for each loop (fo the local). Depending on the tests used: P values, or ROR, should be shown, in the manuscript or in appendices

- Intransitivity: Reviewer #1 has pointed out that you made conclusions on transitivity, that are not supported in the manuscript, To conclude that you need to be clear about how this assumption was evaluated, considering the variables you describe that were used.

- Certainity of the evidence:Reviewer #1 has recommended to apply GRADE to determine what would be the certainity of the evidence (Also called quality of the evidence) with GRADE. We support that recommendation.

- Heterogeneity: Reviewer #1 pointed out as well that you didn't mention anything on the heterogeneity assessment. Heterogeneity should be evaluated and stataisitically assessed: Both: for the whole netwrok, and for pairwise comparisons. Also, in cases of high heterogeneity, special analyses such as Meta-regression and subgroup analyses shpuld have been performed.

- Reviewer #2 has described how your conclusions do not 100% reflect the results.

We expect you could provide appropriate answers to the concerns I have highlighted above along with the reviewers comments which are key to consider your manuscript.

Reviewers' comments:

Reviewer's Responses to Questions

**Comments to the Author**

1. Is the manuscript technically sound, and do the data support the conclusions?

Reviewer #1: Yes

Reviewer #2: Partly

2. Has the statistical analysis been performed appropriately and rigorously? 

Reviewer #1: Yes

Reviewer #2: Yes

3. Have the authors made all data underlying the findings in their manuscript fully available?

Reviewer #1: Yes

Reviewer #2: Yes

4. Is the manuscript presented in an intelligible fashion and written in standard English?

Reviewer #1: Yes

Reviewer #2: Yes

5. Review Comments to the Author

Reviewer #1: Review for First-line targeted therapies of advanced hepatocellular carcinoma: A Bayesian

network analysis of randomized controlled trials

The authors evaluated first-line targeted therapies of advanced hepatocellular carcinoma via a network meta-analysis. This is the first NMA adding important information in the field of advanced hepatocellular carcinoma. However, I have concerns due to the lack of NMA methodological aspects and I think that authors should conduct a lot of things aiming to provide a comprehensive NMA analysis.

My comments focus as follows:

Abstract

I suggest to provide more information in methods section.

Please revise this sentence. “Direct and indirect evidence were 29 combined to time to progress (TTP), overall survival (OS), progress-free 30 survival (PFS), objective response rate (ORR), and the proportion of 31 Grade 3-5 adverse events (G3-5AE) and surface under the cumulative 32 ranking curve (SUCRA) of patients with advanced HCC.”

Provide the measure of outcomes and SUCRA values in a more informative and clear way in the sentence.

Please provide also that you implement the network meta-analysis model and the effect size used (OR and HR).

Materials and methods

Please refer about your NMA protocol. Have you a published protocol of the analysis?

Data synthesis and analysis

I advise authors to start this section reporting that they performed the NMA model and the measures used for treatment effects estimates.

Authors need to provide a global test for the assessment of inconsistency, such as the random-effects design-by-treatment interaction model and the local test (node-splitting). Then, authors could report in results section that it not applicable as they have star NMAs.

Missing details for Bayesian NMA. Authors report about posterior distributions but no further details for assumptions, for example prior distribution for model parameters. What priors for model parameters distributions? What prior distribution for heterogeneity? What assumption about heterogeneity, did they used a common between-study standard deviation across all treatment comparisons in each network? What about the assessment of transitivity? Authors need to compute measures for heterogeneity, for example I squared. Sensitivity analyses need to be conducted when it is needed. Rating the quality of the evidence in the estimates using GRADE criteria can also be conducted. Pairwise meta-analyses can also be conducted and I also suggest to conduct the comparison adjusted funnel plot.

Study characteristics

“There was no evidence that the transitivity assumption was violated in 177 any of the networks.” How authors conclude to this? Authors should refer about transitivity in later section of the analysis.

Results

I recommend to provide the results section taking into account all the methodological suggestions are given in Methods.

Reviewer #2: Authors perfored a network meta-analysis of the available RCTs comparing the efficacy of the various targeted therapes used in the management of HCC. This appeared to be a well conducted study with appropriately evaluted outcomes. I have a few comments:

1) Does the BCLC criteria of the included patients inflence the outcomes especially the overall survival? Is there a scope to perform sub-group analysis matching the patient and tumour charateristics in the current or future studies?

2) Are the conclusions reflecting the results appropriately? Soraferenib appears to remain superior from your analysis. THis should be one of the conclusions.

3)in the eligibility criteria: 'The key inclusion

85 criteria for study populations: more than 4 weeks since most recent local

therapy or no local therapy; no prior systemic therapy.' what does this mean?

6. PLOS authors have the option to publish the peer review history of their article (what does this mean?). If published, this will include your full peer review and any attached files.

Reviewer #1: No

Reviewer #2: No

---

## [Author Response · Author response to Decision Letter 0]

18 Nov 2019

Replies to Reviewers

First of all, we thank both reviewers and editors for your positive and constructive comments and suggestions.

Replies to Reviewer 1: 

1. QUESTION: “Abstract

I suggest to provide more information in methods section.

Please revise this sentence. “Direct and indirect evidence were combined to time to progress (TTP), overall survival (OS), progress-free survival (PFS), objective response rate (ORR), and the proportion of Grade 3-5 adverse events (G3-5AE) and surface under the cumulative ranking curve (SUCRA) of patients with advanced HCC.”

Provide the measure of outcomes and SUCRA values in a more informative and clear way in the sentence.

Please provide also that you implement the network meta-analysis model and the effect size used (OR and HR).”

Answer: We have revised the sentence. It has been replaced with “Time to progress (TTP), overall survival (OS), progress-free survival (PFS), were calculated as hazard ratios (HRs) and 95% credible intervals (CIs). Objective response rate (ORR) and the proportion of Grade 3-5 adverse events (G3-5AE) were expressed as odds ratios (ORs) and 95% CIs. We pooled study-specific HRs and ORs using fixed-effects network meta-analyses, and ranked first-line drugs by the surface under the cumulative ranking curve (SUCRA).”

2. QUESTION: “Materials and methods

Please refer about your NMA protocol. Have you a published protocol of the analysis?”

Answer: We were registered our NMA protocol on July 29, 2019. But the records are being assessed. So, we didn’t have a published protocol.

3. QUESTION: “Data synthesis and analysis

I advise authors to start this section reporting that they performed the NMA model and the measures used for treatment effects estimates.

Authors need to provide a global test for the assessment of inconsistency, such as the random-effects design-by-treatment interaction model and the local test (node-splitting). Then, authors could report in results section that it not applicable as they have star NMAs.

Missing details for Bayesian NMA. Authors report about posterior distributions but no further details for assumptions, for example prior distribution for model parameters. What priors for model parameters distributions? What prior distribution for heterogeneity? What assumption about heterogeneity, did they used a common between-study standard deviation across all treatment comparisons in each network? What about the assessment of transitivity? Authors need to compute measures for heterogeneity, for example I squared. Sensitivity analyses need to be conducted when it is needed. Rating the quality of the evidence in the estimates using GRADE criteria can also be conducted. Pairwise meta-analyses can also be conducted and I also suggest to conduct the comparison adjusted funnel plot.”

Answer: We rewrote this paragraph. We reused R language to perform statistical calculations. We had reported the NMA model and the measures used for treatment effects estimates. We had added the detailed description of Bayesian NMA. We dad provided consistency checks for the assessment of inconsistency. We had added heterogeneity test for both pairing comparison and NMA. The transitivity was assessed by examining the patient baseline characteristics across studies (age, gender, performance status and tumor stage), treatment stage and treatment protocol. The GRADE evaluation system and funnel plots were also added into this NMA.

4. QUESTION: “Study characteristics

‘There was no evidence that the transitivity assumption was violated in any of the networks.’ How authors conclude to this? Authors should refer about transitivity in later section of the analysis.”

Answer: We had added the description of transitivity in later section of the analysis. “To confirm the transitivity and the loop-specific consistency assumption, pairwise direct and indirect effect estimates of closed loops of evidence were inspected for any disagreement. The transitivity was assessed by examining the patient baseline characteristics across studies (age, gender, performance status and tumor stage), treatment stage and treatment protocol.”

5. QUESTION: “Results

I recommend to provide the results section taking into account all the methodological suggestions are given in Methods.”

Answer: New results were provided according to the new calculation.

Replies to Reviewer 2: 

1. QUESTION: “Does the BCLC criteria of the included patients inflence the outcomes especially the overall survival? Is there a scope to perform sub-group analysis matching the patient and tumour charateristics in the current or future studies?”

Answer: The BCLC criteria of the included patients dose influence the outcomes especially the overall survival. However, the included studies didn’t group by the BCLC criteria. So, we could not perform sub-group analysis. Fortunately, the tumour charateristics of included patients were relatively consistent.

2. QUESTION: “Are the conclusions reflecting the results appropriately? Soraferenib appears to remain superior from your analysis. THis should be one of the conclusions.”

Answer: The conclusion had been supplemented by the affirmation of sorafenib. “For the moment, sorafenib was still as a first-line drug of first choice.”

3. QUESTION: “in the eligibility criteria: 'The key inclusion criteria for study populations: more than 4 weeks since most recent local therapy or no local therapy; no prior systemic therapy.' what does this mean?”

Answer: It means that it should last more than 4 weeks for the included study populations since most recent local therapy or no local therapy. It was to avoid the influence of other treatments.

We appreciate for editors/reviewers’ warm work earnestly, and hope that the correction will meet with approval.

Thank you and best regards.

Yours sincerely,

Xuezhong Xu

E-mail: xxzdoctor@163.com.

---

## [Decision Letter · Decision Letter 1]

17 Dec 2019

PONE-D-19-22935R1

First-line targeted therapies of advanced hepatocellular carcinoma: A Bayesian network analysis of randomized controlled trials

PLOS ONE

Dear Mr Xu,

Thank you for submitting your manuscript to PLOS ONE. After careful consideration, we feel that it has merit but does not fully meet PLOS ONE’s publication criteria as it currently stands. Therefore, we invite you to submit a revised version of the manuscript that addresses the points raised during the review process.

We would appreciate receiving your revised manuscript by Jan 31 2020 11:59PM. To enhance the reproducibility of your results, we recommend that if applicable you deposit your laboratory protocols in protocols.io, where a protocol can be assigned its own identifier (DOI) such that it can be cited independently in the future. For instructions see: http://journals.plos.org/plosone/s/submission-guidelines#loc-laboratory-protocols

We look forward to receiving your revised manuscript.

Kind regards,

Ivan D. Florez

Academic Editor

PLOS ONE

Additional Editor Comments (if provided):

Thanks for submitting a revised version. Your revised manuscript has incorporated many of reviewers comments. However, in addition to reviewer #1 comments (which are below) there are a couple of points that hasn’t been completely addressed from reviewer #2, and additional issues identified by myself as Academic Editor:

Reviewer #2 points to be addressed:

1. Reviewer #2 pointed out how the BCLC criteria for the patients could have had an impact on the overall survival. You have pointed out that a subgroup analyses is not possible since the authors didn’t group patients by BCLC criteria. However, there are alternatives to this, considering that you agree that this stage could have an impact on the outcome, and therefore it can be a potential effect modifier.

- If authors provided a proportion of patients with a specific BCLC, you could group studies according to specific proportions

- If a subgroup analyses is not feasible at this point you should address this as a limitation in a paragraph at the end of the discussion section of your manuscript, and discuss there how were the tumor characteristics of the patients and how this may impact on the results.

2. Regarding reviewer#2 comment about the conclusions, you provide the next sentence: “For the moment, sorafenib was still as a first-line drug of first choice.” We think that according to the results, Srafenib should not be highlighted as the best approach. I think In conclusions you should emphasize in those interventions that were superior, BUT adding the quality of evidence for those. See the following sentence as an example:

For OS, Van (100 and 300mg), seem to be the best options with Low and moderate quality of evidence, respectively. For G3 5AE, Van 100 and 300mg), and play seem to be the best interventions, with low quality of evidence all of them (Just as an example, you need to apply GRADE methodology as I suggest below, and find the final quality assessment).

3. Also, regarding this comment from reviewer #2:

In the eligibility criteria: 'The key inclusion criteria for study populations: more than 4 weeks since most recent local therapy or no local therapy; no prior systemic therapy.' what does this mean?” You have provided a response, However, it is not reflected in the paper. This part should be clear enough for reader so they could have the same question as Reviewer #2. Thus, please detail in this section, what does that mean and the reasons for that decision.

Major and minor Comments from Editor:

MAJOR:

You have explained your GRADE approach. However, major issues in your GRADE approach have been identified:

- You state that you used the methods recommended by: Puhan et al (2014). However, using only this method is not appropriate as it was the first approach and it has had at least 2 updates: Brignardello-Petersen et al. J Clin Epidemiol 2018, Brignardello-Petersen et al. J Clin Epidemiol 2019, Consider applying the full current approach to assess the quality of evidence.

- According to GRADE approach (Puhan’s and Brignardello-Petersen’s articles) you need to assess the quality of the direct, the indirect and the network evidence for each comparison and each outcome. Although you may present only your Network estimates and your network quality assessments in your main manuscript. However, you need to provide evidence that you conducted all the process appropriately and for that you need to provide all the direct, indirect and NMA estimates + quality of evidence. We cannot identify in your appendices these estimates (Except by some of them that are provided in the Forest plots, and without each GRADE assessment) for readers that would be interested in them.

- Also in page 35 you state:

“Therefore, in the present NMA, the GRADE score was downgraded universally due to the diversity of patient characteristics and the disunity of evaluation standard”. This sentence is very problematic, because this is not what GRADE guidance recommends. GRADE working group never recommends to downgrade comparisons “globally”. This shows a misunderstanding of the approach. If you apply GRADE, you should follow all the recommended steps by Puhan’s and Brignardello-Petersen’s articles which are the official GRADE working group Guidance.

- in the same page you state:

“Second, it was further downgraded in certain comparisons for no direct evidence”. That may be true. But again, the only way to identify what comparison were downgraded and for what reasons (GRADE Criteria that were use to downgrade in each direct and indirect and NMA estimates)), is to provide in appendices a full list of all the available estimates (D, I, and NMA) for all the outcomes, along with the corresponding GRADE assessment, following the GRADE guidance, and indicating the reasons for downloading each one of them, according to the 7 criteria recommended by the GRADE approach.

- Also, in the same page, you state: “In addition, it was further downgraded in some relevantcomparisons because of their IF below 50%”. Could you explain how this criterion match with the GRADE approach recommended by Puhan’s and Brignardello-Petersen 2018 and 2019, One more time, there should be evidence of what comparisons were downgraded by any particular criterion

- Finally, as you state that you calculated I2. There is no information in the manuscript nor the appendices that shows the i2 for all the direct comparisons. You should provide this data as you described you calculated them to assess the heterogeneity of direct comparisons

MINOR:

Additional minor changes to consider are:

- You excluded one study and the reason provided is that it was data deficient. please clearly explain what does that mean.

- Page 14, line 80, Please write PRISMA in Capital letters.

- Page 14, line 80. There is a specific PRISMA Guidance for NMA, it seems you followed the PRISMA for regular Sr and MA. You need to follow Guidance that is specific for NMA (PRISMA NMA). You also need to provide the appropriate reference for the PRISMA-NMA Guidance.

- Page 14, line 82. Since you cannot provide a PROSPERO registration number at this time, lease indicate the date you submitted the register to the PROSPERO database. Also indicate what was the state of this PROSPERO submission by the time you submitted this manuscript to PlosOne

- FIGURE 4 is of very low quality, you need to provide a better image

- Page 35: They “IF” acronym has not ben explained before and has not been detailed in the methods as a key approach

Reviewers' comments:

Reviewer's Responses to Questions

**Comments to the Author**

1. If the authors have adequately addressed your comments raised in a previous round of review and you feel that this manuscript is now acceptable for publication, you may indicate that here to bypass the “Comments to the Author” section, enter your conflict of interest statement in the “Confidential to Editor” section, and submit your "Accept" recommendation.

Reviewer #1: (No Response)

2. Is the manuscript technically sound, and do the data support the conclusions?

Reviewer #1: Yes

3. Has the statistical analysis been performed appropriately and rigorously? 

Reviewer #1: Yes

4. Have the authors made all data underlying the findings in their manuscript fully available?

Reviewer #1: Yes

5. Is the manuscript presented in an intelligible fashion and written in standard English?

Reviewer #1: Yes

6. Review Comments to the Author

Reviewer #1: I am happy that the authors provided a registered protocol in PROSPERO database. I recommend authors to revise the sentence “It did not have an online protocol…..under review” with “The protocol has been registered in PROSPERO database and it is now under review”

The authors addressed most of the concerns and suggestions about the NMA methodology and the manuscript has been improved. Although, I have some comments and concerns for inconsistency checking.

Based on review comments, authors should have provided a global test for inconsistency assumption, eg. Design-by-treatment interaction model (DBT) and if inconsistency was detected they should have provided a node-splitting approach.

I don’t agree with the interpretation provided for the results of the Node-splitting method. Node-splitting approach checks if direct and indirect is in agreement (consistency). Authors concluded to “The result showed a robust and homogeneous network of evidence”. This should be revised. The results of all tests (P-values) for each outcome could be reported in the manuscript with S7, S13, S16 and S19 figures.

All the forest plots provided for NMA should be renamed in figures and tables with Comparison-adjusted funnel plot. Moreover, dashed lines in plots are missing and should be provided (Figure S20).

7. PLOS authors have the option to publish the peer review history of their article (what does this mean?). If published, this will include your full peer review and any attached files.

Reviewer #1: No

---

## [Author Response · Author response to Decision Letter 1]

14 Jan 2020

Dear Editors and Reviewers:

Thank you very much for your comments and suggestions.

We have revised the manuscript, according to the comments and suggestions of reviewers and editor, and responded, point by point to, the comments as listed below.

I would like to re-submit this revised manuscript to PLOS ONE, and hope it is acceptable for publication in the journal.

Looking forward to hearing from you soon.

With kindest regards,

Yours Sincerely 

Xuezhong Xu.

Replies to Editors

1. QUESTION: Reviewer #2 pointed out how the BCLC criteria for the patients could have had an impact on the overall survival. You have pointed out that a subgroup analyses is not possible since the authors didn’t group patients by BCLC criteria. However, there are alternatives to this, considering that you agree that this stage could have an impact on the outcome, and therefore it can be a potential effect modifier. 

- If authors provided a proportion of patients with a specific BCLC, you could group studies according to specific proportions.

- If a subgroup analyses is not feasible at this point you should address this as a limitation in a paragraph at the end of the discussion section of your manuscript, and discuss there how were the tumor characteristics of the patients and how this may impact on the results. 

ANSWER: We thank very much for the editor's advice. However, the proportions of patients with different BCLC stages were not uniform. So, the subgroup analysis is not feasible. So, we added some discussion at the end of the discussion section. “The distributions of BCLC stages in different studies were not in full accord. Patients with B or C stage often had worse prognosis than those with A stage. The BCLC criteria for the patients could have an impact on the overall survival. Fortunately, the vast majority of patients include in this analysis were in stage B or C.”

2. QUESTION: Regarding reviewer#2 comment about the conclusions, you provide the next sentence: “For the moment, sorafenib was still as a first-line drug of first choice.” We think that according to the results, Srafenib should not be highlighted as the best approach. I think In conclusions you should emphasize in those interventions that were superior, BUT adding the quality of evidence for those. See the following sentence as an example: For OS, Van (100 and 300mg), seem to be the best options with Low and moderate quality of evidence, respectively. For G3 5AE, Van 100 and 300mg), and play seem to be the best interventions, with low quality of evidence all of them (Just as an example, you need to apply GRADE methodology as I suggest below, and find the final quality assessment).

ANSWER: We fully endorse the editor's suggestion. We had revised the sentence with “For OS, Van (100 and 300mg), seem to be the best options with low and moderate quality of evidence, respectively. For G3-5AE, Van (100 and 300mg), seem to be the best interventions, with low and very low quality of evidence all of them.”

3. QUESTION: Also, regarding this comment from reviewer #2: In the eligibility criteria: 'The key inclusion criteria for study populations: more than 4 weeks since most recent local therapy or no local therapy; no prior systemic therapy.' what does this mean?” You have provided a response, However, it is not reflected in the paper. This part should be clear enough for reader so they could have the same question as Reviewer #2. Thus, please detail in this section, what does that mean and the reasons for that decision.

ANSWER: We had revised the sentence with “To avoid the influence of other treatments, the key inclusion criteria for included study populations were as follows: First, it should last more than 4 weeks since most recent local therapy or no local therapy. Second, the patients did not receive prior systemic therapy.”

4. QUESTION: - You state that you used the methods recommended by: Puhan et al (2014). However, using only this method is not appropriate as it was the first approach and it has had at least 2 updates: Brignardello-Petersen et al. J Clin Epidemiol 2018, Brignardello-Petersen et al. J Clin Epidemiol 2019, Consider applying the full current approach to assess the quality of evidence.

- According to GRADE approach (Puhan’s and Brignardello-Petersen’s articles) you need to assess the quality of the direct, the indirect and the network evidence for each comparison and each outcome. Although you may present only your Network estimates and your network quality assessments in your main manuscript. However, you need to provide evidence that you conducted all the process appropriately and for that you need to provide all the direct, indirect and NMA estimates + quality of evidence. We cannot identify in your appendices these estimates (Except by some of them that are provided in the Forest plots, and without each GRADE assessment) for readers that would be interested in them.

- Also in page 35 you state:“Therefore, in the present NMA, the GRADE score was downgraded universally due to the diversity of patient characteristics and the disunity of evaluation standard”. This sentence is very problematic, because this is not what GRADE guidance recommends. GRADE working group never recommends to downgrade comparisons “globally”. This shows a misunderstanding of the approach. If you apply GRADE, you should follow all the recommended steps by Puhan’s and Brignardello-Petersen’s articles which are the official GRADE working group Guidance.

- in the same page you state:“Second, it was further downgraded in certain comparisons for no direct evidence”. That may be true. But again, the only way to identify what comparison were downgraded and for what reasons (GRADE Criteria that were use to downgrade in each direct and indirect and NMA estimates)), is to provide in appendices a full list of all the available estimates (D, I, and NMA) for all the outcomes, along with the corresponding GRADE assessment, following the GRADE guidance, and indicating the reasons for downloading each one of them, according to the 7 criteria recommended by the GRADE approach.

- Also, in the same page, you state: “In addition, it was further downgraded in some relevant comparisons because of their IF below 50%”. Could you explain how this criterion match with the GRADE approach recommended by Puhan’s and Brignardello-Petersen 2018 and 2019, One more time, there should be evidence of what comparisons were downgraded by any particular criterion.

- Finally, as you state that you calculated I2. There is no information in the manuscript nor the appendices that shows the i2 for all the direct comparisons. You should provide this data as you described you calculated them to assess the heterogeneity of direct comparisons

ANSWER: Thanks very much to the editors for pointing out my shortcomings. We looked through the literature “Brignardello-Petersen R, Bonner A, Alexander PE, Siemieniuk RA, Furukawa TA, Rochwerg B, et al. Advances in the GRADE approach to rate the certainty in estimates from a network meta-analysis. J Clin Epidemiol 2018; 93:36–44. https://doi.org/10.1016/j.jclinepi.2017.10.005 PMID: 29051107”. We have relearned the GRADE approach and rewritten this paragraph. The method was described as follows “For direct comparison, we graded evidence from the five aspects; risk of bias, inconsistency, indirectness, imprecision and publication bias, using the standard GRADE approach. For indirect comparison, we rated evidence according to the lower grades of direct comparisons and intransitivity. For NMA estimates, we rated evidence according to the higher grades of the direct and indirect comparisons and incoherence.” The direct, indirect, and NMA Estimates for OS with the GRADE Assessment were shown in S22-23 Tables. The I2 for all the direct comparisons were shown in S5 Figure, S9 Figure, S11 Figure, S14 Figure and S17 Figure, and they were also shown in S22-23 Tables. We have deleted the “IF” criterion.

5. QUESTION: - You excluded one study and the reason provided is that it was data deficient. please clearly explain what does that mean.

- Page 14, line 80, Please write PRISMA in Capital letters.

- Page 14, line 80. There is a specific PRISMA Guidance for NMA, it seems you followed the PRISMA for regular Sr and MA. You need to follow Guidance that is specific for NMA (PRISMA NMA). You also need to provide the appropriate reference for the PRISMA-NMA Guidance.

- Page 14, line 82. Since you cannot provide a PROSPERO registration number at this time, lease indicate the date you submitted the register to the PROSPERO database. Also indicate what was the state of this PROSPERO submission by the time you submitted this manuscript to PlosOne

- FIGURE 4 is of very low quality, you need to provide a better image

- Page 35: They “IF” acronym has not been explained before and has not been detailed in the methods as a key approach

ANSWER: - The excluded study was lack of control group.

- Page 14, line 80, we have revised PRISMA in Capital letters. We have rewritten the list following the guidance of PRISMA-NMA checklist.

- Page 14, line 82. This network meta-analysis has been registered in the PROSPERO public database (CRD42019145188)

- FIGURE 4 had been deleted and been replaced by S22-23 Tables.

- Page 35: The “IF” acronym means “information fraction”. It was used to assess statistical power and strength of evidence for each treatment comparison. However, we decide to remove this assessment because the GRADE approach was enough.

Replies to Reviewers

First of all, we thank both reviewers and editors for your positive and constructive comments and suggestions.

Replies to Reviewer 1: 

1. QUESTION: Based on review comments, authors should have provided a global test for inconsistency assumption, eg. Design-by-treatment interaction model (DBT) and if inconsistency was detected they should have provided a node-splitting approach.

Answer: Thanks very much for the advice. We did lack a check for global consistency. However, design-by-treatment interaction model (DBT) was a frequentist NMA model that considers both heterogeneity between studies and inconsistency between study designs according to White IR (White IR. Network meta-analysis. Stata J 2015; 15: 951-985.). Since what I used was bayesian model, design-by-treatment interaction model (DBT) was not suitable for me. In Konstantinos’s study (Konstantinos K, Panagiotis K, Stavros S, et al. Comparative effectiveness of different transarterial embolization therapies alone or in combination with local ablative or adjuvant systemic treatments for unresectable hepatocellular carcinoma: A network meta-analysis of randomized controlled trials[J]. PLOS ONE, 2017, 12(9): e0184597), the author used unrelated mean effects model to evaluate the inconsistency. So, we also used this method to estimate the inconsistency. And the modification has been added in the revised manuscript. The results of comparisons in both consistency and inconsistency models were roughly consistent. The results were shown in S20 Table.

2. QUESTION: I don’t agree with the interpretation provided for the results of the Node-splitting method. Node-splitting approach checks if direct and indirect is in agreement (consistency). Authors concluded to “The result showed a robust and homogeneous network of evidence”. This should be revised. The results of all tests (P-values) for each outcome could be reported in the manuscript with S7, S13, S16 and S19 figures.

Answer: The sentence has been revised with “The node-splitting approach also showed a good consistency between the direct and indirect comparisons”. The results of all tests (P-values) for each outcome has been added in the manuscript.

3. QUESTION: All the forest plots provided for NMA should be renamed in figures and tables with Comparison-adjusted funnel plot. Moreover, dashed lines in plots are missing and should be provided (Figure S20). 

Answer: In S21 Figure, the titles have been revised as Comparison-adjusted funnel plot. And we have added the dashed lines.

We appreciate for editors/reviewers’ warm work earnestly, and hope that the correction will meet with approval.

Thank you and best regards.

Yours sincerely,

Xuezhong Xu

E-mail: xxzdoctor@163.com.

---

## [Editor Report · Decision Letter 2]

10 Feb 2020

First-line targeted therapies of advanced hepatocellular carcinoma: A Bayesian network analysis of randomized controlled trials

PONE-D-19-22935R2

Dear Dr. Xu,

We are pleased to inform you that your manuscript has been judged scientifically suitable for publication and will be formally accepted for publication once it complies with all outstanding technical requirements.

With kind regards,

Peter Starkel, M.D., Ph.D.

Academic Editor

PLOS ONE

Additional Editor Comments (optional):

All comments have been addressed. No further remarks.
---

## [Editor Report · Acceptance letter]

12 Feb 2020

PONE-D-19-22935R2 

First-line targ veted therapies of advanced hepatocellular carcinoma: A Bayesian network analysis of randomized controlled trials 

Dear Dr. Xu:

I am pleased to inform you that your manuscript has been deemed suitable for publication in PLOS ONE. Congratulations! Your manuscript is now with our production department. 

With kind regards,

on behalf of

Dr Peter Starkel 

Academic Editor

PLOS ONE